# An improved particle swarm optimization for multilevel thresholding medical image segmentation

Jiaqi Ma, Jianmin Hu*

GBA Branch of Aerospace Information Research Institute, Chinese Academy of Sciences, Guangzhou, Guangdong province, China

* hujm@aircas.ac.cn

**Data Availability Statement:** The manuscript already contains most of the data, and the remaining data is stored in the public repository: https://github.com/JiaQiMark/CIWP-PSO/tree/main/result.

## Abstract

Multilevel thresholding image segmentation is one of the widely used image segmentation methods, and it is also an important means of medical image preprocessing. Replacing the original costly exhaustive search approach, swarm intelligent optimization algorithms are recently used to determine the optimal thresholds for medical image, and medical images tend to have higher bit depth. Aiming at the drawbacks of premature convergence of existing optimization algorithms for high-bit depth image segmentation, this paper presents a pyramid particle swarm optimization based on complementary inertia weights (CIWP-PSO), and the Kapur entropy is employed as the optimization objective. Firstly, according to the fitness value, the particle swarm is divided into three-layer structure. To accommodate the larger search range caused by higher bit depth, the particles in the layer with the worst fitness value are employed random opposition learning strategy. Secondly, a pair of complementary inertia weights are introduced to balance the capability of exploitation and exploration. In the part of experiments, this paper used nine high-bit depth benchmark images to test the CIWP-PSO effectiveness. Then, a group of Brain Magnetic Resonance Imaging (MRI) images with 12-bit depth are utilized to validate the advantages of CIWP-PSO compared with other segmentation algorithms based on other optimization algorithms. According to the segmentation experimental results, thresholds optimized by CIWP-PSO could achieve higher Kapur entropy, and the multi-level thresholding segmentation algorithm based on CIWP-PSO outperforms the similar algorithms in high-bit depth image segmentation. Besides, we used image segmentation quality metrics to evaluate the impact of different segmentation algorithms on images, and the experimental results show that the MRI images segmented by the CIWP-PSO has achieved the best fitness value more times than images segmented by other comparison algorithm in terms of Structured Similarity Index and Feature Similarity Index, which explains that the images segmented by CIWP-PSO has higher image quality.

**Funding:** The author(s) received no specific funding for this work.

**Competing interests:** The authors have declared that no competing interests exist.

# 1 Introduction

As an important technology in the field of image and vision, medical image segmentation has brought many significant contributions to modern society [1,2]. In the past decades, image segmentation has been widely used in cell detection [3], medical diagnose [4], autonomous driving [5], remote sensing [6,7], etc. Segmentation of image is a foundational process with numerous medical applications, including functional image and medical analysis, and segmentation of image can be regarded as an upstream task of medical image processing, which prepares for downstream image analysis. The principle of the image segmentation is to merge pixels with uniform attribute into several regions and extract the areas of interest [8]. Generally speaking, image segmentation technology can be roughly divided into two categories. One is traditional image segmentation technology based on thresholds, regions, and edges, and the other is convolutional neural networks (CNN) based on deep learning.

With the development of deep learning technology and computing power in recent years, CNN can handle a variety of image segmentation problems very well [9,10]. To obtain higher solution accuracy, CNN relies too much on a large amount of labeled data. However, manual calibration of datasets for image segmentation is time-consuming and labor-intensive, and what's even more terrible is that for a considerable number of rare diseases, the medical community does not have enough data for neural network training. Therefore, traditional image segmentation methods are particularly important in the field of medical image segmentation.

The traditional medical image segmentation methods mainly include thresholding method, region growing method and edge-based method. Among them, thresholding image segmentation has the advantages of fast processing speed, stable performance and simple operation, so it is widely used in the field of medical image segmentation [11]. The principle of thresholding image segmentation method is to classify pixel points based on the preset thresholds and set pixels in the same category to the same gray value. Thresholding image segmentation method can be divided into two categories, binary level thresholding and multi-level thresholding. Bi-level thresholding segmentation simply uses a threshold to separate the image into background and foreground, and multilevel thresholding segmentation segment the image into $n+1$ groups of pixels based on n thresholds [12]. In comparison, multilevel threshold image segmentation technology can adapt to a variety of tasks and handle more complex images.

In the process of multilevel threshold image segmentation, the determination of threshold is the most critical step. The exhaustive search method once is used to search the optimal thresholds, but it has a large computational burden in determining the optimal thresholds, and its computational burden increases exponentially as the number of thresholds increases [13]. To improve the search efficiency of the optimal thresholds, swarm intelligent optimization algorithms (SIOA) are introduced by many scholars to search for the optimal threshold [14–16]. The SIOA greatly reduce the computational burden of searching for optimal thresholds, which promotes the application of multilevel threshold image segmentation in medical image processing.

Different from the 8-bit depth images seen in daily life, the images presented by medical imaging machines often have a higher bit (12-bit) depth to obtain more texture features and physiological information. However, the current mainstream SIOA used for image threshold segmentation mostly search for optimal thresholds on 8-bit depth images. Compared with 8-bit deep images, the search range of 12-bit deep images is 16 times larger for the optimization algorithm, so the current SIOA perform poorly on thresholds searching for high-bit depth images, exposing the disadvantage of premature convergence. Therefore, this paper proposed an improved particle swarm optimization (CIWP-PSO) for the multilevel thresholds search of high-bit depth images.

Particle swarm optimization (PSO) takes into account both the present and history, local and global aspects, and has great potential to search for optimal values in a large range. The

key to converging to the optimal threshold in a large range is to avoid falling into the local optimum for a long time. Therefore, this paper adjusts the particle swarm to a pyramid structure based on the fitness value, and the worst particle group adopts a random opposition learning strategy to jump out of the local range. In addition, a pair of complementary inertia weights related to fitness value are introduced, which can balance the local and global capability with a greater extend to adapt to a larger search range.

In the part of experiments, 9 12-bit depth benchmark images and 12-bit depth brain tumor MRI images are selected by this paper, and many SIOA and improved PSO are compared with CIWP-PSO to validate the superiority of the CIWP-PSO. The experiments results show that CIWP-PSO can achieve more excellent Kapur entropy, and the images segmented by CIWP-PSO's thresholds have high image quality in terms of Peak Signal-to-Noise ratio (PSNR) and Structured Similarity Index Measure (SSIM).

The main contributions of this paper are presented as follows:

- This paper proposes a new pairwise complementary inertia weight strategy to balance exploration and development to a greater extent.

- The random learning strategy are introduced into the PSO to adapt to the optimal threshold search for high-bit depth medical images.

- This paper carried out lots of experiments to evaluate the segmentation of the CIWP-PSO and its peers using benchmark and MRI medical images with various metrics.

The rest of this paper is organized as follows. Section 2 discusses the relevant literature and research in the field of medical image segmentation. Section 3 briefly introduces the principle of initial PSO. The proposed CIWP-PSO and its improved strategies are explained in detail in Section 4. Section 5 employed many image segmentation experiments to evaluate the segmentation quality of CIWP-PSO and other SIOA. Finally, the conclusions and the future directions of medical image thresholding segmentation are given in Section 6.

## 2 Related works

This section will discuss related work from the perspectives of swarm intelligent algorithm and thresholding image segmentation.

### 2.1 Swarm intelligent optimization algorithm

The SIOA has strong practicality and is simple to operate and easy to deploy. With the development of information technology and computing power, SIOA intelligent algorithm has become increasingly popular in solving optimization problems [17,18]. Sati et al [19] combined the swarm intelligent optimization algorithm and PID algorithm to automatically control the power system. Mustaqeem [20] et al used the swarm intelligent optimization algorithm and multilayer perception to optimize software defect prediction.

Thanks to SIOA's strong scalability, SIOA can be improved to adapt to different optimization problems. Therefore, in recent decades, various improvement strategies have been introduced into different SIOA, solving many optimization problems very well [21]. Zhang et al [22] applied strengthening swarm hierarchy strategy to gray wolf optimization (GWO) to optimize the parameters of multilayer perceptron for the cancer identification. Their improved GWO algorithm focuses on solving high-dimensional optimization problems but is not suitable for optimization problems with larger search ranges. Gong et al [23] used diversity migration strategy to improve the particle swarm optimization algorithm to improve the diversity of population, which greatly improved the exploration of the algorithm. Shasha Yang et al [24]

utilized sparrow search algorithm (SSA) improved by hybrid strategy to optimize the hyper-parameters of Long-short term memory network for geological disasters forecasting. The SSA has the advantages of fast convergence speed and high convergence accuracy and can obtain high-precision solutions within a short number of iterations. However, the SSA pays too much attention on the local areas and ignores the ability of exploration. Ant colony algorithm and A* search algorithm are combined by Yanfei Zhang [25] to plan the ship route. Yanchun Gu [26] uses self-developed inverted S-shaped inertia weights to improve chicken swarm optimization for high-dimensional problems. The inverted S-shaped inertia weight takes into account the impact of the number of iterations on flock convergence and attempts to balance exploitation and exploration by the number of iterations. However, the inverted S-shaped inertia weight strategy relies on thousands of long-term iterations, which greatly increase the computational burden and is not suitable for processing image data.

## 2.2 Multilevel thresholding image segmentation

Multilevel thresholding image segmentation has the characteristics of simple operation and high execution efficiency and can segmentation an image into multiple regions by relying only on a set of thresholds. The adaptive thresholding image segmentation combined with SIOA greatly improves the search efficiency of optimal thresholds. Compared with deep neural networks, adaptive thresholding image segmentation does not rely on massive data, which makes it still receive widespread attention at a time when neural network algorithms are sweeping the field of image processing.

Chunzhi Wang et al [27] utilized a mixed-strategy-based improved whale optimization algorithm (WOA) for multilevel thresholding image segmentation. They proposed k-point search strategy to randomly generate an initialization population with a more uniform distribution, and adaptive weight coefficient was introduced to balance exploration and exploitation, which makes the improved WOA achieve more excellent Kapur entropy and Otsu value. Bald eagle search (BES) was improved by Sharma et al [28] to segmentation brain MRI images. They introduced the dynamic opposite learning strategy to overcome the shortcomings of slow convergence and local optima stagnation of original BES. Lin Lan, and Shengsheng Wang [29] used the improved African vulture optimization algorithm (AVOA) based on predation memory strategy to segment chest X-ray images and brain MRI images. The predation memory strategy speeds up convergence and improved the solution accuracy, and their experiment results indicate that the medical image segmented by the improved AVOA has higher image quality in many evaluation metrics, such as Jaccard Similarity Coefficient, SSIM and Feature Similarity Index (FSIM). However, all the above threshold optimization methods do not focus on high-bit depth medical image threshold search, and they are prone to falling into local optima when faced with a wide range of objective functions.

## 3 Method

### 3.1 Particle swarm optimization

Inspired by the flocking behaviors of birds, PSO was propose by James Kennedy and Russell Eberhart in 1995 [30]. As a method of evolutionary computation, PSO has been widely used in many fields for the advantages of simplicity and high effectiveness [31]. Thanks to its scalability, PSO has become increasingly popular in recent years [32,33].

The PSO algorithm simulates the process of bird foraging, in which the position of the particle is updated by its own best historical position and the best position of the swarm. In the evolution procedure, each particle iteratively learns from its own experience and exchanges

information with the best one of swarm. Based on the above principles, the optimization procedure of PSO is as follows.

1. Swarm initialization: Randomly generate a swarm containing N particles each with position vector $x_i = (x_{i,1}, x_{i,2}, \ldots, x_{i,D})$, and then randomly generate a velocity vector $v_i = (v_{i,1}, v_{i,2}, \ldots, v_{i,D})$ with dimension D for each particle.

2. Swarm evaluation: Calculate the fitness value of each particle, and then record the historical optimal position $p_{best}$ of each particle and the optimal individual $g_{best}$ of the population.

3. Position update: In each iteration, particle update its velocity vector and position vector according to Eqs (1) and (2) respectively.

$$v_i(t+1) = \omega v_i(t) + c_1 r_1 [p_{best} - x_i(t)] + c_2 r_2 [g_{best} - x_i(t)] \tag{1}$$

$$x_i(t+1) = x_i(t) + v_i(t) \tag{2}$$

where $\omega$ is the inertia weight, usually between 0.4 and 0.9; $c_1$ and $c_2$ represents the learning factors of oneself and group respectively; $r_1$ and $r_2$ takes the random numbers uniformly distributed between 0 and 1; $t$ is the number of iterations [34].

4. After each position iteration, update the $p_{best}$ and $g_{best}$.

5. If the conditions for the end of iteration is met, the optimal solution is returned, and the algorithm ends; otherwise, go back to Step (3).

## 3.2 System model and objective function

This subsection describes the system model and problem definition of adaptive threshold segmentation in detail. Thresholding image segmentation divides pixels into multiple regions based on thresholds. Assuming that there is a medical image $I$ with $L$ gray levels, and we need use $n$ thresholds $th_1, th_2, th_3, \ldots, th_n$ to segment the image into $n+1$ regions, which can be described as Eq (3).

$$C_0 = \{I(i,j) \in I | 0 \leq I(i,j) \leq th_1 - 1\},$$

$$C_1 = \{I(i,j) \in I | th_1 \leq I(i,j) \leq th_2 - 1\},$$

$$C_2 = \{I(i,j) \in I | th_2 \leq I(i,j) \leq th_3 - 1\},$$

$$\cdots$$

$$C_n = \{I(i,j) \in I | th_n \leq I(i,j) \leq L\} \tag{3}$$

where $i$ and $j$ represent the coordinates of pixels on the image; $C_i$ is the $i$th set of pixels. When $n = 1$, there is only one threshold, also known as bi-level thresholding segmentation; while when $n \geq 2$, these n thresholds divide the image into n+1 regions according to the grayscale value of the pixel. This paper mainly focusses on multilevel thresholding segmentation, whose $n$ is greater than 1.

The purpose of the multilevel thresholding segmentation is to segment pixels with similar gray values into the same region as much as possible, which can ignore the secondary information and highlight the important contour information of the image. Therefore, we need an objective

function as the criterion to evaluate the quality of the thresholds, and the objective function can not only judge whether the pixels between each threshold have reached the maximum degree of separation, but also guide the evolution of the optimization algorithm. Assuming that $f(I, th_1, th_2, \ldots, th_n)$ is the objective function for thresholding image segmentation to evaluate the quality of segmented image. The goal of the adaptive thresholding segmentation algorithm is to search the optimal thresholds to get the most excellent solution of objective function.

In the field of threshold image segmentation, there are two evaluation criteria mostly commonly used as objective function, namely Otsu method and Kapur entropy [35–37]. Otsu's maximum inter-class variance method is a commonly used threshold segmentation method. Its basic idea is to find an optimal threshold by iteratively calculating the inter-class variance under different thresholds so that the inter-class variance between foreground and background pixels is maximized. Otsu's maximum inter-class variance method can automatically determine the optimal threshold to avoid the difficulty of manual selection and can achieve good segmentation results for images of different gray levels. Kapur entropy is also a threshold selection method based on information theory. Its basic idea is to select the optimal threshold by calculating the entropy of pixels with different gray levels in the image. In an image, the gray value of a pixel can be regarded as a random variable, and the distribution of pixel values can reflect the information content of the image. Kapur entropy finds an optimal threshold by calculating the entropy of pixels with different gray levels, so that the information difference between the foreground and background pixels after segmentation is maximized.

The Otsu method aims to the maximum inter-class variance of the grayscale values of different segmented regions, and the inter-class variance tends to be maximum when the areas of each region are similar. Therefore, the Otsu method is good at processing images with similar areas of different segmented regions, and it is easy to segment images with targets of different sizes into regions with similar sizes. However, Kapur entropy method aims to maximize the entropy, which is not sensitive to the size of the divided regions based on the gray-level intra-class probability and can better retain small targets in the image. Therefore, this paper selected the Kapur entropy as the optimization objective of CIWP-PSO. Kapur entropy [38] is known as the maximum entropy method, and the function of the Kapur entropy of the image can be formulated as Eq (4).

$$f_{Kapur}(t_1, t_2, t_3, \ldots, t_n) = H_0 + H_1 + H_2 + H_3 + \ldots + H_n$$

$$H_0 = -\sum_{i=0}^{t_1-1} \frac{p_i}{\omega_0} \ln \frac{p_i}{\omega_0}, \omega_0 = \sum_{i=0}^{t_1-1} p_i$$

$$H_1 = -\sum_{i=t_1}^{t_2-1} \frac{p_i}{\omega_1} \ln \frac{p_i}{\omega_1}, \omega_1 = \sum_{i=t_1}^{t_2-1} p_i$$

$$H_2 = -\sum_{i=t_2}^{t_3-1} \frac{p_i}{\omega_2} \ln \frac{p_i}{\omega_2}, \omega_2 = \sum_{i=t_2}^{t_3-1} p_i$$

$$\ldots$$

$$H_n = -\sum_{i=t_n}^{L-1} \frac{p_i}{\omega_n} \ln \frac{p_i}{\omega_n}, \omega_n = \sum_{i=t_n}^{L-1} p_i \tag{4}$$

where the $t_1$, $t_2$, $t_3$,. . .,$t_n$ segmented the image into $n+1$ regions; $p_i$ represents the probability of $i$th gray level and $\sum_{i=0}^{L-1} p_i = 1$; $L$ is the maximum value of gray level.

## 3.3 The improved CIWP-PSO

This subsection proposes improved CIWP-PSO based on two strategies and provides a detailed introduction to these two improvement strategies.

**3.3.1 Random opposition learning strategy based on pyramid structure.** Random opposition learning strategy (ROL) is an optimization strategy that uses reverse thinking to increase flexibility, and it is applied in many optimization algorithms [39,40]. The original intension of ROL is to escape from local optimality and improve the population diversity. The core idea of this strategy is that if a solution performs poorly, its opposite solution may have better potential. While exploring opposite points in the search space, adding a little random element will increase the diversity of the population, which greatly increases the ability of global exploration and increases the probability of escaping the local optimum. By introducing random opposite solutions, the solution space can be explored more comprehensively, thereby accelerating convergence and improving global search capabilities. In SIOA, for the proportion of individuals with the worst fitness, the weak guidance of the excellent individuals is not enough to make them escape from the area with poor fitness value, while ROL can powerfully help these individuals jump out of local areas, which is great for the searching over large areas.

ROL needs to act on a proportion of poor individuals, but original PSO does not divide the population into obvious hierarchies. Therefore, this paper divides the particle swarm into a three-layer pyramid structure based on the fitness value, and it is shown as Fig 1.

The population is divided into three layers from top to bottom according to the fitness value from good to bad. As shown in Fig 1, most individuals are divided into the second layers, and the first layer contains only one global optimal individual, and the third layer only contains 5% ~ 20% of the worst individuals. The global best one serving as the first layer is used to guide the update of the second layer individuals. A certain proportion of individuals with the worst fitness values are set as the third layer of the pyramid, and the remaining individuals serve as the second layer.

Thanks to the three-layer hierarchical structure, ROL can act on the individuals in the third layer, who have the worst fitness value, so these poor individuals can explore other positions to obtain better fitness values and help the population escape from the local optima. The particles in the third layer use ROL as the position update strategy, The ROL can be formulated as Eq (5).

$$P(t + 1) = LB + UB - r*P(t) \tag{5}$$

where $t$ represents the number of iterations; $LB$ and $UB$ are the upper and lower boundaries of search range respectively; $r$ is a uniformly distributed random number between 0 and 1.

**3.3.2 Complementary inertia weights.** In the population evolution process of all swarm optimization algorithms, the mathematical model of individual position update can be uniformly expressed by the Eq (6):

$$P_{t+1} = P_t + \Delta p \tag{6}$$

where $P_t$ and $P_{t+1}$ represent the position vector of the current iteration and the next iteration respectively, while $\Delta p$ represents the distance that the individual should move at the current iteration. In the initial PSO, although the update process of each particle's velocity vector has a

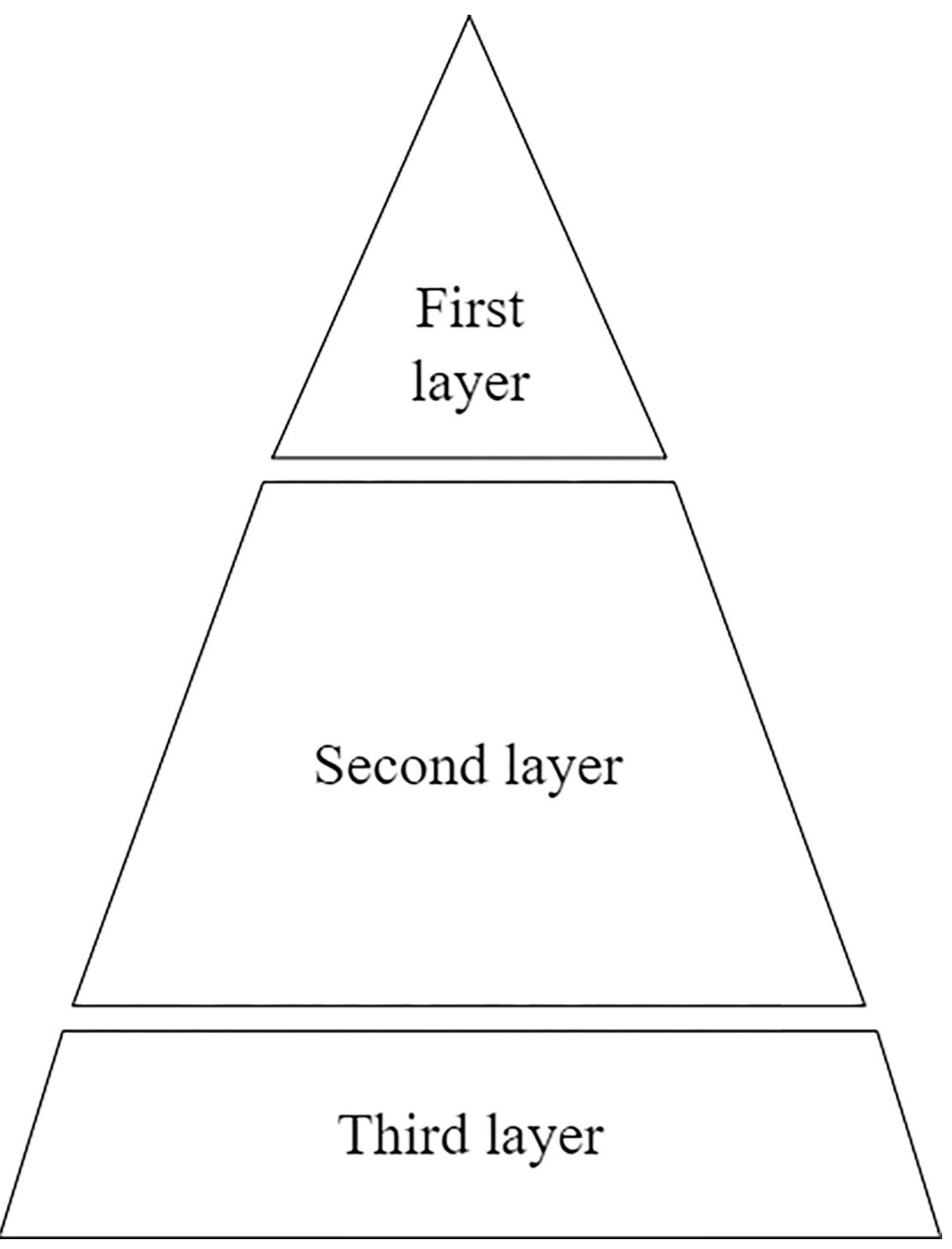

**Fig 1. The three-layer pyramid structure of particle swarm.**

weight parameter as shown Eq (1), the weight parameter $\omega$ cannot adapt to the changes in fitness value and iteration times, which greatly limits the performance of the algorithm.

Based on the above mathematical model of position update, many scholars have adopted various inertia weight mechanisms to adjust $P_t$ or $\Delta p$ to improve the convergence efficiency and convergence accuracy of the optimization algorithm [41,42]. However, the traditional inertia weights are only used to adjust the step size $\Delta p$, ignoring the impact of initial position vector $P_t$ on the position update process, which greatly reduces the guidance of inertia weights and limits the convergence ability of optimization algorithms.

In order to address the limitation of traditional inertia weights, the pair of complementary inertia weights proposed by this paper uses the inverse relationship between the initial position vector and step size according to the fitness value to conveniently control both initial position vector $P_t$ and the step length $\Delta p$. The more excellent fitness value makes the particle has greater motivation to stay in its original position and the poor fitness value makes the particle has more motivation to leave current position (The larger fitness is, the more excellent fitness is). Therefore, the inertia weight $\omega(f)$ increases as the fitness increases, and reversed inertia weight $\omega_{reverse}(f)$ decreases as the fitness increases. The mechanism of the pair of inertia weights is based on the mathematical model of position update described as below.

$$P_{t+1} = \omega(f) \cdot P_t + \omega_{reverse}(f) \cdot \Delta p \tag{7}$$

where inertia weight $\omega(f)$ and reverse inertia weight make up this pair of complementary inertia weights. In this position update model, when the more excellent fitness value makes the particle has greater motivation to stay in its original position, so it should get a larger inertia weight and smaller reverse weight. When the particle has poor fitness value, the particle has more motivation to leave current position, so it should get a larger reverse weight and smaller inertia weight. Besides, to guide each particle to search in a large range more accurately, the factor that controls the step size $\Delta p$ also needs to be designed, and reverse weight can also play the role of adjusting the step size. Inertia weight and reverse weight need to have opposite trends to form a complementary dynamic, and the changing trend of the pair of inertia weights in this paper needs to meet the following requirements:

- The slope of the function should be monotonic throughout the variation range, such that the inertia weight is positively related to fitness and reverse weight is negatively related to fitness.

- Keeping the slope larger at the beginning and end to ensure that more weights are distributed around the median which can amplify the influence of inertia weight to accommodate search over a wide range.

The inverse Sigmoid function satisfies these two requirements for the pair of inertia weights. The slope of the inverse Sigmoid function is monotonic throughout the variation range, such that the inertia weight $\omega(f)$ is positively related to fitness and reverse weight $\omega_{reverse}(f)$ is negatively related to fitness. The inverse Sigmoid function has the larger slope at the beginning and end to ensure that more weights are distributed around the median, which can amplify the influence of inertia weight to accommodate search over a wide range.

Therefore, this paper selects inverse Sigmoid function as the basis function, and the inverse Sigmoid function is described as Eq (8).

$$S_{inverse} = (\log(s_x \cdot x + m_x)/(m_x - s_x \cdot x))/s_y \tag{8}$$

where $s_x$ and $s_y$ represent the scaling factors in the x-axis and y-axis directions, which are chosen 0.96 and 7.8 respectively. $m_x$ is the movement factor in the x-axis direction, which is set to 0.5. Based on the above inverse Sigmoid function, the inertia weight and reverse weight proposed by this paper can be formulated as Eqs (9) and (10).

$$\omega(f) = (\omega_{min} + \omega_{max})/2 + (w_{max} - w_{min}) \cdot S_{inverse}(x) \tag{9}$$

$$\omega_{reverse}(f) = (\omega_{min} + \omega_{max})/2 - (w_{max} - w_{min}) \cdot S_{inverse}(x) \tag{10}$$

where $\omega_{min}$ and $\omega_{max}$ represent the minimum and maximum of inertia weight respectively; $f$ is the fitness value of current particle, and $f_{best}$ and $f_{worst}$ represent the best and worst fitness

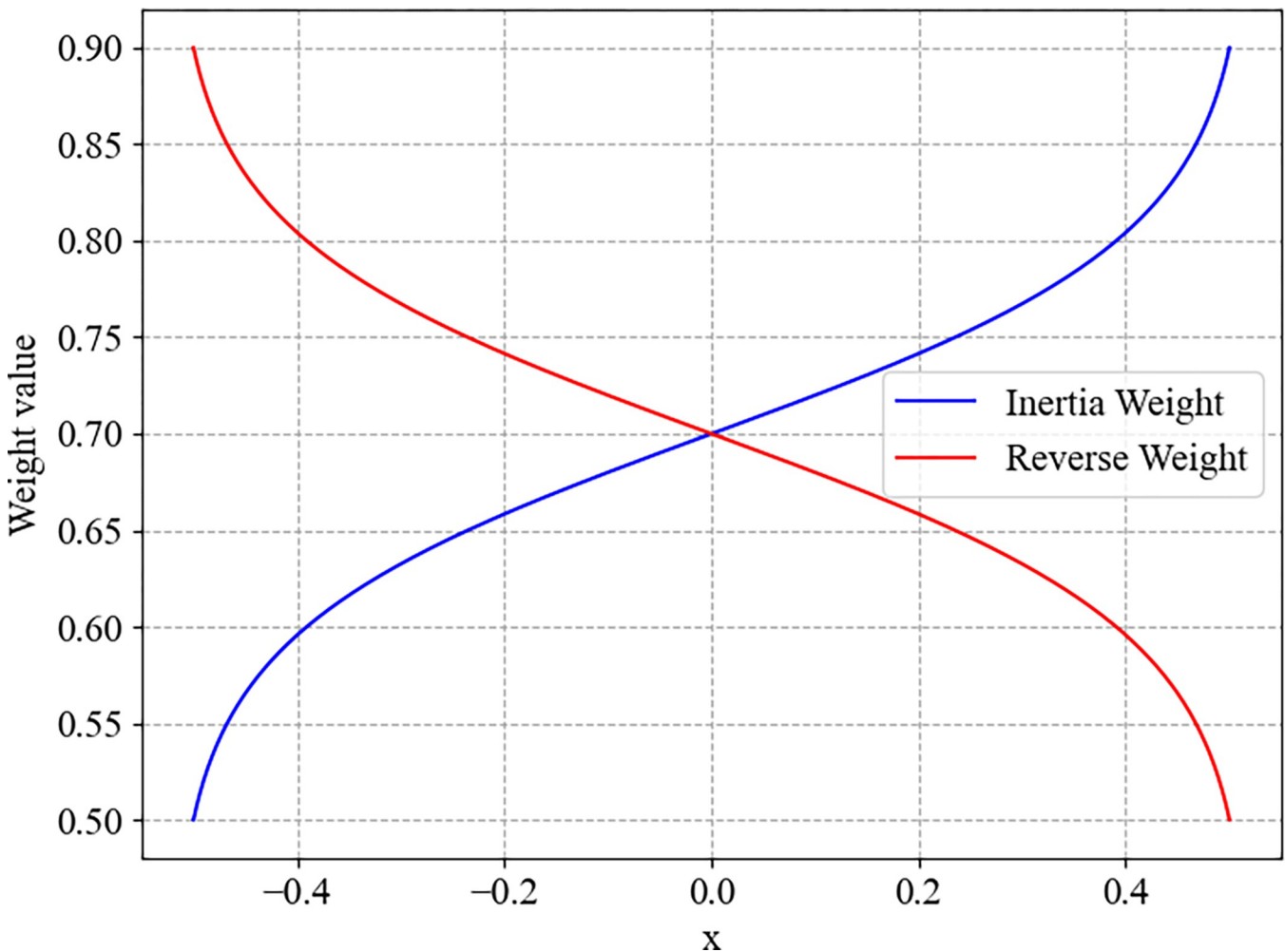

**Fig 2. The curve of inertia weight ($w_{min}$ = 0.5, $w_{max}$ = 0.9).**

respectively. x can be described as Eq (11).

$$x = (f - (f_{best} + f_{worst})/2)/(f_{best} - f_{worst}) \tag{11}$$

The position update equation of PSO modified by the pair of complementary inertia weight is shown in Eqs (12) and (13).

$$v_i(t+1) = \omega(f) \cdot v_i(t) + \omega_{reverse}(f) \cdot (c_1 r_1 [p_{best} - x_i(t)] + c_2 r_2 [g_{best} - x_i(t)]) \tag{12}$$

$$x_i(t+1) = x_i(t) + v_i(t) \tag{13}$$

The curve of the inertia weight and reverse weight are shown in Fig 2.

From Fig 2, we can intuitively see that when the fitness value of a particle is greater than the median fitness, a larger inertia weight and a smaller reversed weight can be obtained to maintain its own initial advantage; when the particle's fitness value is poor, the inertia weight takes a smaller value and the reversed weight takes a greater value to try to get rid of its own position.

**3.3.3 The procedure of CIWP-PSO.** This paper first divides the particle swarm into a three-layer pyramid structure, and then uses the above two strategies to improve PSO. The framework of CIWP-PSO is shown in Algorithm 1.

```
Algorithm 1. The pseudo-code of CIWP-PSO.
a) Input: Set the population size N, dimension D, the maximum number
of iterations T
b) Output: The best position and corresponding fitness
c) Initialize the position vector and velocity vector of each
particle.
d) While (t≤T)
e)     Calculate the fitness value of each particle.
f)     Divide them into three layers based on fitness value.
g)     For p in second layer
h)         Update their position and velocity according to Eqs (12) and
(13)
i)     For p in third layer
j)         Update their position according to Eq (5)
k)     t = t+1
l) Return the best position and corresponding fitness.
```

The overall impact of improvement strategies proposed by this paper on the CIWP-PSO in optimizing thresholding image segmentation is mainly reflected in the following two aspects. Random opposite learning strategy helps the CIWP-PSO give up exploring the worst areas and instead explore more potential thresholds, increase population diversity, and escape from the local optimal thresholds. The pair of complementary inertia weights, using the reverse changes of step size and inertia, provide CIWP-PSO with efficient guidance and feedback on its journey to explore the optimal thresholds, which greatly improves the efficiency of searching for the optimal thresholds in high bit-depth images.

## 4 Experiments

In the section, this paper carries out two main sets of experiments to demonstrate the advantages of the CIWP-PSO in solving the thresholding segmentation problem for high-bit depth image. The first experiment used general benchmark images with high-bit depth to preliminary validate the effectiveness of the algorithm. In the second one, high-bit depth medical images segmented by CIWP-PSO are compared with other SIOAs.

All above experiments were conducted on a person computer with Intel(R) Core (TM) i7-9700 CPU, 16GB RAM and Windows 11 operating system, and the algorithms were implemented with python 3.10.9.

### 4.1 Comparative experiments on benchmark images

**4.1.1 Dataset preprocessing.** In this work, we collected a dataset containing 16 color images with 8-bit depth, and these 16 grayscale images were used as benchmark images to prove the algorithm's segmentation capability for high-bit-depth images. After converting the color images into grayscale images, we mapped the gray values of the gray images to 12-bit depth. The processed images and their grayscale histogram are listed in Fig 3.

**4.1.2 Experiments and result analysis.** To investigate the performance of the CIWP-PSO, six automatic thresholding segmentation algorithms based on widely used swarm intelligent algorithms (GWO, WOA, SSA, PSO, GA [43] and [FA] [44]) are compared to segment the above benchmark images to obtain higher Kapor entropy. In addition, two well-behaved PSO

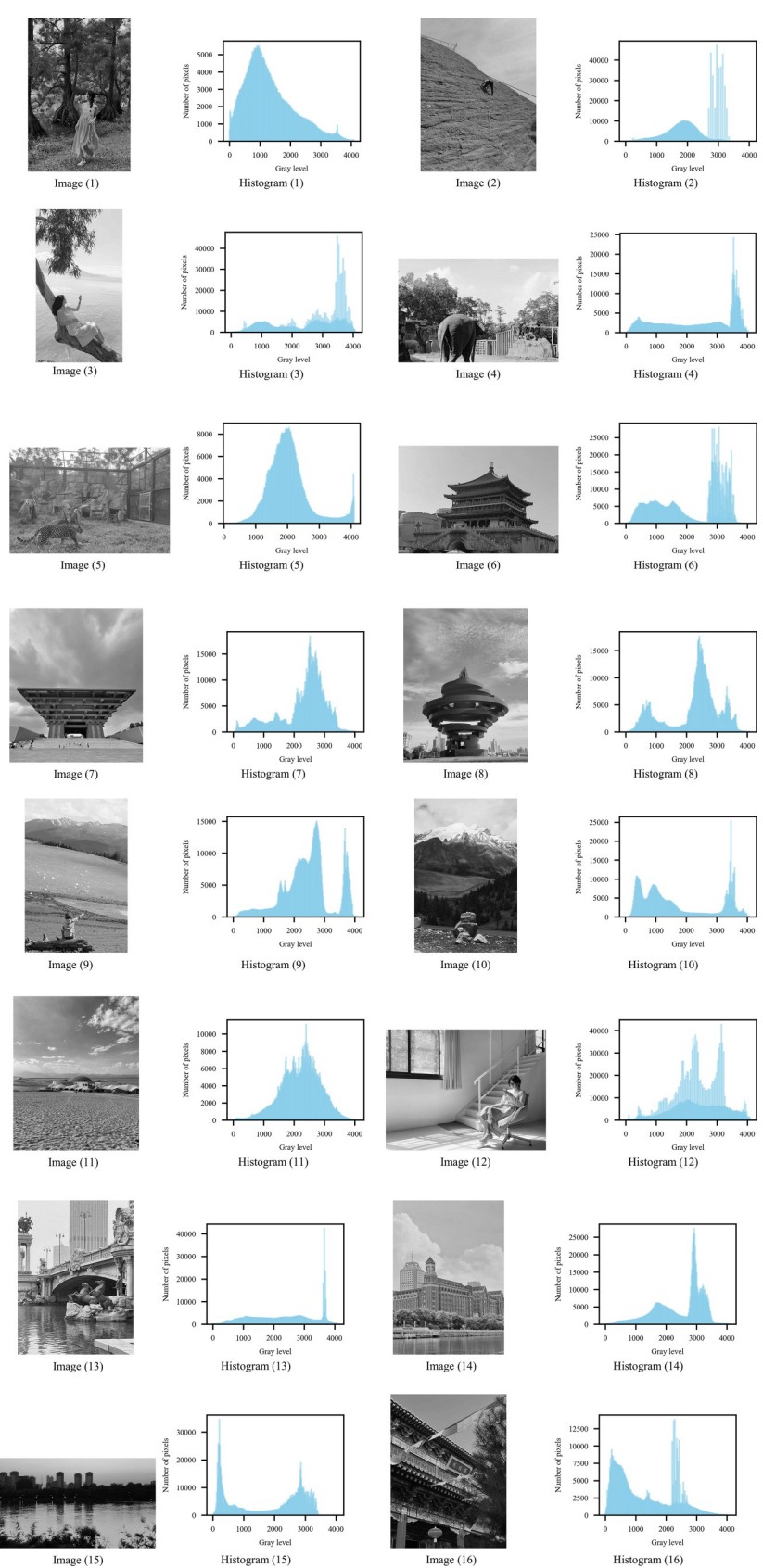

**Fig 3. Grayscale images with 12-bit depth and their corresponding histogram.** Image (1) Histogram (1) Image (2) Histogram (2) Image (3) Histogram (3) Image (4) Histogram (4) Image (5) Histogram (5) Image (6) Histogram (6) Image (7) Histogram (7) Image (8) Histogram (8) Image (9) Histogram (9) Image (10) Histogram (10) Image (11) Histogram (11) Image (12) Histogram (12) Image (13) Histogram (13) Image (14) Histogram (14) Image (15) Histogram (15) Image (16) Histogram (16).

variants, including QPSO [23] with inertia weight strategy and PSOH [45] which combines the strengths of SSA and PSO, are selected to highlight the efficiency of the improved strategies proposed by this paper.

All the optimization algorithms have an initial population size of 30 and a maximum number of iterations of 100. Other hyperparameters for these algorithms are chosen from their initial paper and are presented in Table 1.

To comprehensively compare the multi-thresholding segmentation capability of each algorithm, these benchmark images are segmented by different number of thresholds nTh = [4,6,8] respectively. All algorithms were run independently 10 times, and the middle value were taken to eliminate chance. All the above algorithms regard Kapur entropy as the fitness function, and the Kapur entropies obtained by segmenting the nine benchmark images are listed in Table 2, in which the most excellent Kapur entropy are in bold and blue.

The experimental results form Table 2 shows that the PSO, QPSO, PSOH and CIWP-PSO usually converge to higher fitness value than other SIOAs, which means that the PSO-based algorithms have greater potential to handle the segmentation task of high-bit-depth images, and this is also one of the important reasons why this paper selects PSO as the basic algorithm. Among all algorithms, CIWP-PSO proposed by this paper is the most outstanding one. For almost all segmentation tasks of all images, CIWP-PSO can always search for the relative optimal thresholds to obtain the best Kapur entropy.

To intuitively demonstrate the convergence characteristics of CIWP-PSO, we show the convergence curves of the better-performing algorithms for image (2) and image (9) in Fig 4.

It can be seen from the Fig 4 that CIWP-PSO have significantly smoother convergence curves than original PSO and WOA, indicating that CIWP-PSO has stronger stability. In summary, the multilevel thresholding algorithm based on CIWP-PSO can obtain the most excellent segmentation results for benchmark images with high-bit depth. Apart from this, during the iteration process, CIW converges stably and has good robustness.

**Table 1. Hyperparameters of the above comparison algorithm.**

| Algorithm | Hyperparameters |
|---|---|
| GWO | $r_1 = rand(0,1)$, $r^2 = rand(0,1)$, $a \in [0,2]$ |
| WOA | $b = 0.5$, $a_{min} = 0$, $a_{max} = 2$ |
| SSA | $ST = 0.8$, $P_{producer} = 0.2$, $P_{aware} = 0.1$ |
| PSO | $c_1 = 1.5$, $c_2 = 1.5$, $\omega = 0.8$ |
| GA | $P_{crossover} = 0.35$, $P_{mutation} = 0.1$ |
| FA | $\alpha = 1.0$, $\beta_{min} = 1.0$, $\gamma = 0.01$ |
| QPSO | $c_1 = 1.2$, $c_2 = 1.2$ |
| PSOH | $c_1 = 1.5$, $c_2 = 1.5$, $\omega = 0.8$, $ST = 0.8$ $P_{producer} = 0.2$ $P_{aware} = 0.1$ |
| CIWP-PSO | $c_1 = 1.5$, $c_2 = 1.5$ $w_{min} = 0.5$ $w_{max} = 0.9$, $rate = 0.9$ |

**Table 2. The Kapur entropy obtained by algorithms over the 16 benchmark images with 12-bit depth.**

| Image id | nTh | GWO | WOA | SSA | PSO | GA | FA | QPSO | PSOH | CIWP-PSO |
|---|---|---|---|---|---|---|---|---|---|---|
| Image (1) | 4 | 19.253 | 19.258 | 19.2602 | **19.2648** | 19.2138 | 19.2646 | **19.2648** | **19.2648** | **19.2648** |
| | 6 | 24.5055 | 24.638 | 23.9808 | 24.7973 | 24.5618 | 24.7859 | **24.7991** | 24.7977 | 24.7649 |
| | 8 | 29.2532 | 29.3396 | 28.8557 | 29.7289 | 29.465 | 29.7229 | 29.7643 | 29.684 | **29.7686** |
| Image (2) | 4 | 17.4345 | 17.437 | 16.8056 | **17.4656** | 17.0658 | 17.4539 | **17.4656** | 17.4244 | **17.4656** |
| | 6 | 22.4739 | 22.3895 | 21.4848 | 22.5377 | 22.0004 | 22.4711 | 22.5025 | 22.4342 | **22.5483** |
| | 8 | 26.6023 | 26.2393 | 25.0013 | **27.2971** | 26.0451 | 27.0313 | 27.2056 | 26.4428 | 27.2059 |
| Image (3) | 4 | 18.5586 | 18.5454 | 18.0493 | **18.5604** | 18.3462 | 18.5567 | **18.5604** | **18.5604** | **18.5604** |
| | 6 | 23.5979 | 23.6578 | 23.6803 | 23.8289 | 23.1354 | 23.8025 | 23.7762 | 23.7364 | **23.8324** |
| | 8 | 28.1419 | 28.2484 | 27.3824 | 28.5587 | 28.0036 | 28.568 | 28.6257 | 28.3676 | **28.7423** |
| Image (4) | 4 | 19.0601 | 19.072 | 18.7946 | 19.075 | 18.5146 | 19.077 | **19.0771** | 19.0768 | **19.0771** |
| | 6 | 24.318 | 24.4345 | 24.4875 | 24.5336 | 24.3551 | 24.5018 | 24.5336 | 24.5298 | **24.5372** |
| | 8 | 28.8973 | 29.0716 | 28.8425 | 29.3124 | 28.8445 | 29.2552 | **29.3195** | 29.2929 | 29.1532 |
| Image (5) | 4 | 18.2265 | 18.3074 | 18.2789 | **18.3092** | 18.272 | 18.3082 | 18.3088 | 18.3059 | **18.3092** |
| | 6 | 23.4263 | 23.5921 | 23.049 | **23.6953** | 22.9654 | 23.6768 | 23.6802 | 23.6898 | 23.6855 |
| | 8 | 28.0565 | 28.3972 | 27.9515 | 28.489 | 27.8459 | 28.4848 | 28.5083 | 28.5148 | **28.5259** |
| Image (6) | 4 | 18.1017 | 18.2391 | 18.0804 | **18.2516** | 18.1181 | 18.2377 | **18.2516** | 18.2436 | **18.2516** |
| | 6 | 23.1478 | 23.4681 | 22.496 | **23.6669** | 23.0124 | 23.6024 | 23.3013 | 23.6478 | 23.5369 |
| | 8 | 27.3323 | 28.1239 | 27.6357 | 27.8718 | 28.2484 | 28.4494 | 28.3565 | 28.4315 | **28.5337** |
| Image (7) | 4 | 18.7721 | 18.861 | 18.7516 | 18.8655 | 18.4312 | 18.854 | **18.8663** | 18.8621 | **18.8663** |
| | 6 | 24.2425 | 24.3202 | 23.88 | 24.3848 | 23.7698 | 24.3747 | 24.3994 | 24.3974 | **24.4029** |
| | 8 | 28.5599 | 28.7419 | 28.6786 | 29.0484 | 28.4469 | 29.255 | **29.3163** | 28.9922 | 29.2052 |
| Image (8) | 4 | 18.316 | 18.6148 | 18.2677 | **18.6336** | 18.27 | 18.6266 | **18.6336** | 18.4949 | **18.6336** |
| | 6 | 23.4959 | 23.697 | 23.4241 | 23.972 | 23.7224 | 23.9504 | 23.894 | 23.8604 | **23.9861** |
| | 8 | 28.1317 | 28.7224 | 27.6563 | 28.7183 | 28.0367 | 28.7701 | **28.8661** | 28.8444 | 28.859 |
| Image (9) | 4 | 18.3066 | 18.616 | 18.2439 | 18.6203 | 17.9432 | 18.6193 | **18.6204** | 18.6176 | **18.6204** |
| | 6 | 23.661 | 24.2313 | 23.1777 | 24.2612 | 24.0979 | 24.2252 | 24.2891 | 24.0959 | **24.2897** |
| | 8 | 27.8629 | 28.251 | 28.2173 | 28.8428 | 27.9384 | 28.9796 | 29.0295 | 28.9706 | **29.0847** |
| Image (10) | 4 | 18.6903 | 18.7245 | 18.5134 | 18.7418 | 18.2935 | 18.741 | **18.7419** | 18.7405 | **18.7419** |
| | 6 | 23.8911 | 24.0499 | 23.7531 | 24.1714 | 23.6303 | 24.0988 | **24.1716** | 24.1238 | 24.171 |
| | 8 | 28.0086 | 28.8064 | 26.9977 | 28.9975 | 28.2025 | 28.9292 | 29.0382 | 28.7585 | **29.0394** |
| Image (11) | 4 | 18.6715 | 18.761 | 18.2863 | **18.8055** | 18.6567 | 18.804 | 18.7702 | 18.8034 | **18.8055** |
| | 6 | 23.9887 | 24.2161 | 23.8558 | 24.3287 | 23.9896 | 24.3287 | 24.3504 | 24.3512 | **24.3526** |
| | 8 | 28.6289 | 29.0719 | 28.7293 | 29.2135 | 29.0616 | 29.2712 | 29.2866 | 29.262 | **29.3022** |
| Image (12) | 4 | 18.5393 | 18.7374 | 18.691 | 18.7438 | 18.6014 | 18.7442 | **18.7454** | 18.7374 | **18.7454** |
| | 6 | 23.6718 | 23.7419 | 23.6794 | 23.8403 | 23.4517 | 23.8322 | 23.8102 | 23.8571 | **23.8645** |
| | 8 | 27.7621 | 28.4357 | 28.4815 | 28.657 | 27.8965 | 28.4772 | 28.2981 | 28.329 | **28.7221** |
| Image (13) | 4 | 18.4063 | 18.4738 | 18.2103 | 18.4804 | 18.4232 | 18.4787 | 18.4801 | 18.4797 | **18.4805** |
| | 6 | 23.6792 | 23.7915 | 23.47 | 23.7745 | 23.2706 | 23.8148 | 23.8309 | 23.7681 | **23.8654** |
| | 8 | 28.1499 | 28.0849 | 27.2499 | 28.8345 | 28.0154 | 28.7238 | 28.7792 | 28.8126 | **28.8678** |
| Image (14) | 4 | 18.4525 | 18.8619 | 18.3746 | 18.8628 | 18.6329 | 18.8532 | 18.8382 | 18.8023 | **18.8629** |
| | 6 | 23.1689 | 24.1581 | 23.7308 | 24.4324 | 23.873 | 24.4027 | 24.4267 | 24.331 | **24.458** |
| | 8 | 28.8381 | 29.2196 | 28.402 | 29.2558 | 28.051 | 29.3137 | **29.4066** | 29.2511 | 29.405 |
| Image (15) | 4 | 18.2989 | 18.3223 | 18.2607 | **18.3239** | 18.0803 | 18.3155 | **18.3239** | 18.2622 | **18.3239** |
| | 6 | 23.3824 | 23.7814 | 22.6488 | 23.8537 | 23.208 | 23.788 | 23.8686 | 23.879 | **23.8807** |
| | 8 | 27.9095 | 28.1218 | 27.9965 | 28.6711 | 28.0682 | 28.6045 | 28.7148 | 28.4722 | **28.7161** |
| Image (16) | 4 | 18.6888 | 18.683 | 18.5055 | **18.6889** | 18.4954 | 18.6847 | **18.6889** | 18.6863 | **18.6889** |
| | 6 | 23.9869 | 24.1024 | 24.1087 | 24.2183 | 23.9767 | 24.1943 | 24.23 | 24.1703 | **24.2301** |
| | 8 | 28.4036 | 28.8749 | 28.8226 | 29.051 | 28.7741 | 29.1067 | 29.0721 | 28.8101 | **29.2301** |

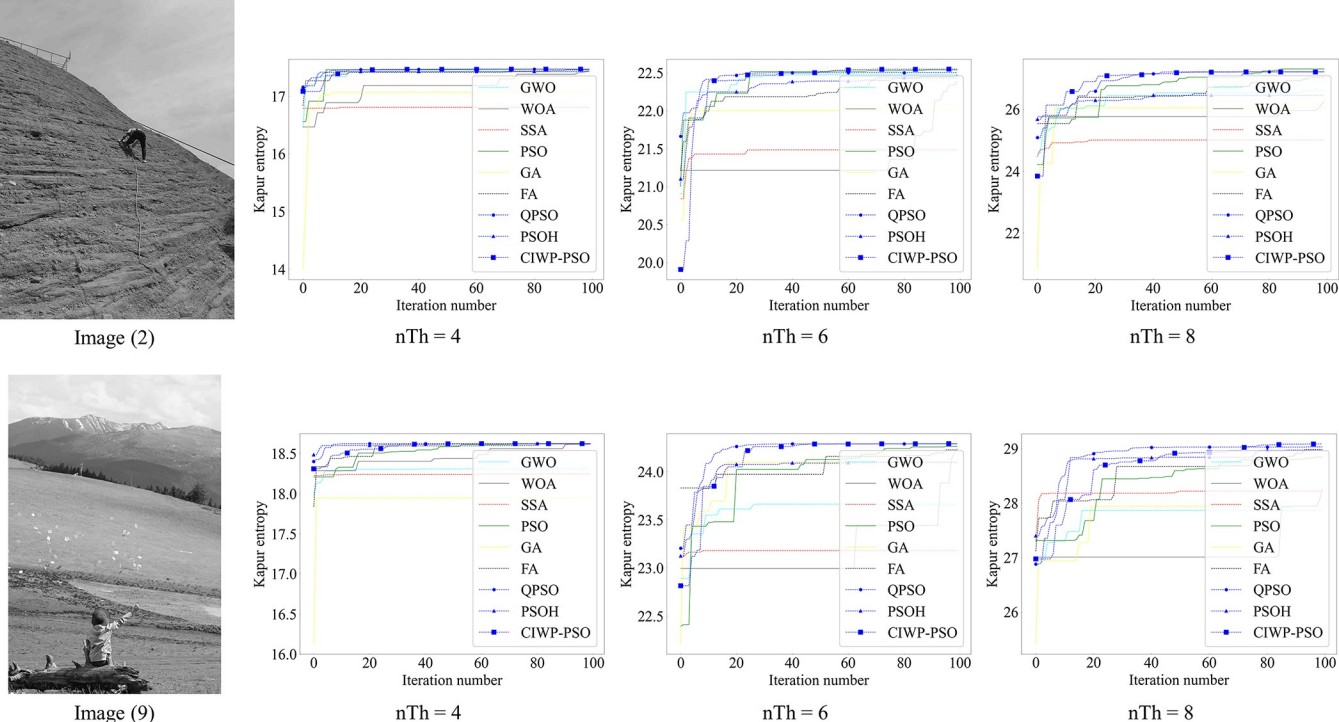

**Fig 4. The convergence curve of the algorithms for image (2) and image (9).** Image (2) nTh = 4 nTh = 6 nTh = 8 Image (9) nTh = 4 nTh = 6 nTh = 8.

## 4.2 Comparative experiments on medical images

In this work, the MRI brain tumor dataset, which are grayscale images with 12-bit depth, are used to test the segmentation capabilities of these algorithms for high-bit depth medical image. From this data set, we selected 9 images as experimental data to comprehensively compare the performance of algorithms, and these 9 images and their corresponding histograms are shown in Fig 5. The set of experiments first demonstrate the ability to segment images, and then evaluate the segmentation impact of CIWP-PSO on the brain tumor images.

**4.2.1 Analysis of experimental results.** In the experiment, GWO, WOA, SSA, PSO, GA, FA, QPSO and PSOH are applied in the images segmentation to compare performance with CIWP-PSO. All above algorithms pursue higher Kapur entropy as the optimization goal, and higher Kapur entropy means better segmentation results. To comprehensively evaluate the multi-thresholding segmentation ability of each algorithm, the MRI images are segmented by different number of thresholds nTh = [4,6,8] respectively. In order to rule out serendipity, all algorithms were run independently 10 times, and the medians of the 10 results of algorithms for each image are presented in Table 3.

From the Table 3, we can get that the Kapur Entropy of all tumor images will increase as the number of thresholds increases. Similar to processing benchmark images, the PSO-based algorithms show remarkable advantages in segmenting high-bit-depth medical images. Overall, the CIWP-PSO obtains the best result in 18 cases (27 in total) which shows the best performance in solution accuracy, whereas the other algorithms show weak advantages in any case when compared with the CIWP-PSO.

**4.2.2 Friedman Rank-sum test.** To prove the validity of experimental data, it is often necessary to introduce non-parametric statistical inference testing methods, and common non-parametric testing methods include Friedman Rank Sum method and Wilcoxon Rank Sum

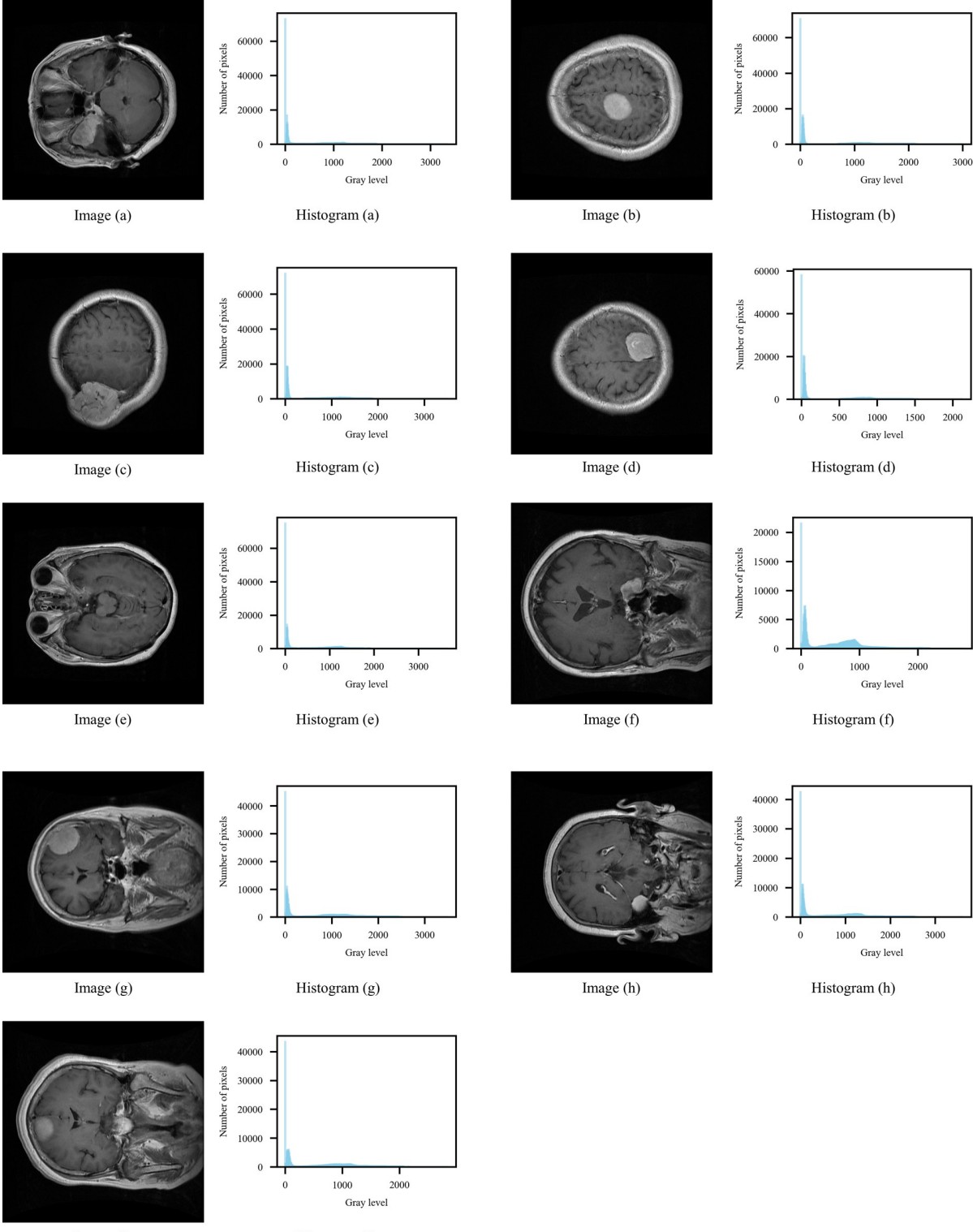

**Fig 5. Images of brain tumors used in the experiments and their corresponding histograms.** Image (a) Histogram (a) Image (b) Histogram (b) Image (c) Histogram (c) Image (d) Histogram (d) Image (e) Histogram (e) Image (f) Histogram (f) Image (g) Histogram (g) Image (h) Histogram (h) Image (i) Histogram (i).

**Table 3. The fitness values (Kapur entropy) obtained by the above algorithms.**

| Image id | nTh | GWO | WOA | SSA | PSO | GA | FA | QPSO | PSOH | CIWP-PSO |
|---|---|---|---|---|---|---|---|---|---|---|
| Image a | 4 | 28.8526 | 28.9529 | 28.892 | 28.9278 | 28.7314 | 28.9541 | 28.8372 | 28.955 | **28.9558** |
| | 6 | 38.5452 | 39.4726 | 38.731 | 39.5246 | 39.3496 | 39.6419 | **39.6761** | 39.6509 | 39.5794 |
| | 8 | 48.9602 | 48.9863 | 48.575 | 49.6855 | 48.2923 | 49.6574 | 49.6023 | 49.5954 | **49.7089** |
| Image b | 4 | 28.2933 | 28.7414 | 27.8398 | **28.7852** | 28.5478 | 28.7745 | 28.7413 | 28.7813 | **28.7852** |
| | 6 | 38.2879 | 39.0177 | 37.4522 | 39.1449 | 38.0368 | **39.188** | 39.1564 | 39.1563 | 39.1343 |
| | 8 | 47.4757 | 48.9497 | 48.2478 | 48.8609 | 47.5649 | 48.8452 | 49.1102 | 48.9905 | **49.1231** |
| Image c | 4 | 28.7669 | 29.1591 | 28.9383 | 29.1776 | 28.7367 | 29.1695 | **29.1778** | 29.1705 | **29.1778** |
| | 6 | 39.1816 | 39.786 | 39.2533 | 39.9822 | 39.0169 | 39.9434 | 39.9458 | **39.9933** | 39.9917 |
| | 8 | 49.4891 | 49.5101 | 48.6948 | 49.9807 | 48.3729 | 49.9433 | 50.0601 | 49.9587 | **50.0902** |
| Image d | 4 | 26.7475 | 27.3979 | 25.7192 | 27.4075 | 26.8845 | 27.3766 | **27.4081** | 27.3989 | 27.4067 |
| | 6 | 36.0916 | 37.0784 | 36.5904 | 37.2134 | 36.1556 | 37.0716 | 37.2098 | 37.1877 | **37.2159** |
| | 8 | 45.1746 | 46.1122 | 40.7018 | 46.1324 | 44.9182 | 45.8545 | 46.3018 | 45.9858 | **46.4223** |
| Image e | 4 | 29.2168 | 29.2937 | 29.244 | 29.2959 | 29.0435 | 29.293 | 29.2957 | 29.2948 | **29.2967** |
| | 6 | 39.2094 | 39.9737 | 38.4882 | 40.0398 | 39.5375 | 40.0207 | **40.0612** | 40.0058 | 40.0045 |
| | 8 | 48.4343 | 50.0213 | 45.9791 | 50.1839 | 49.7253 | 50.1525 | 49.8672 | 50.1893 | **50.1997** |
| Image f | 4 | 29.2852 | 29.3818 | 28.8593 | 29.3867 | 28.8757 | 29.3799 | 29.3498 | 29.3853 | **29.3871** |
| | 6 | 39.1243 | 39.288 | 38.3894 | 39.3356 | 38.8668 | 39.293 | **39.3481** | 39.339 | 39.3392 |
| | 8 | 48.0323 | 43.9579 | 46.771 | 48.6675 | 48.0278 | 48.747 | 48.6379 | 48.8998 | **49.0636** |
| Image g | 4 | 29.4077 | 29.4824 | 28.7575 | 29.4967 | 29.2431 | 29.476 | **29.4973** | 29.4895 | **29.4973** |
| | 6 | 39.5853 | 39.9839 | 39.1342 | 40.1848 | 39.5956 | 40.12 | 40.1816 | **40.1858** | 40.1851 |
| | 8 | 48.8233 | 49.9339 | 47.2116 | 49.8371 | 48.8032 | 49.9474 | 50.0849 | 50.0123 | **50.1022** |
| Image h | 4 | 29.8018 | 30.0989 | 30.0862 | 30.1068 | 29.898 | 30.1035 | **30.1082** | 30.007 | 30.1076 |
| | 6 | 40.1544 | 40.6746 | 39.9839 | 40.7449 | 39.845 | 40.698 | 40.7319 | 40.691 | **40.7511** |
| | 8 | 49.4527 | 50.4936 | 49.0214 | 50.8421 | 49.6848 | 50.7354 | 50.8956 | 50.9198 | **50.9613** |
| Image i | 4 | 28.5991 | 28.9556 | 28.7725 | 28.957 | 28.6052 | 28.9566 | 28.9631 | 28.9631 | **28.9641** |
| | 6 | 38.6742 | 39.0503 | 37.8233 | 39.147 | 38.134 | 39.088 | **39.1514** | 39.1471 | 39.1485 |
| | 8 | 47.7414 | 48.55 | 44.7868 | 48.8982 | 47.529 | 48.597 | 48.8952 | 48.8169 | **48.9058** |

method. In contrast to the Wilcoxon Rank Sum Test, which compares two groups, the Friedman test is designed for scenarios involving three or more related groups. The Friedman test is an extension of the Wilcoxon test, making it versatile for situations where multiple related groups are involved, providing a comprehensive analysis of differences in medians. It involves ranking observations within each group, calculating average ranks, and computing the Friedman statistic [46]. The Friedman test is particularly valuable when comparing the performance of multiple algorithms.

In order to ensure that the differences in experimental data are statistically significant, we calculate the Friedman statistic and p-value of PSO and its variants for the 9 medical images, which are listed in Table 4.

The p-value of these four sets of data is far less than the significance level (0.05), which means that the CIWP-PSO proposed by this paper does have a significant difference between CIWP-PSO and its counterparts. In addition, the average ranks of the above four PSO-based

**Table 4. The Friedman statistic and p-value of PSO and its variants.**

| Algorithms | Statistic | p-value |
|---|---|---|
| PSO, QPSO, PSOH, CIWP-PSO | 19.8102 | 1.8626E-04 |

**Table 5. The average ranks.**

| Algorithms | Average rank | Rank |
|------------|:------------:|:----:|
| PSO | 2.99 | 4 |
| QPSO | 2.13 | 2 |
| PSOH | 2.97 | 3 |
| CIWP-PSO | 1.57 | 1 |

algorithms are presented in Table 5, which clearly shows that the CIWP-PSO is indeed superior to the other two PSO variants in segmenting high-bit depth medical images.

**4.2.3 Image segmentation quality evaluation.** To fully validate the performance of these algorithms, in addition to verifying their ability to segment high-bit depth medical images, this paper also evaluates the impact of segmentation on images using the algorithms that performed well in the above results, including SSA, PSO, GA, QPSO, PSOH and CIWP-PSO. We use Peak Signal-to-Noise ratio (PSNR), Structured Similarity Index Measure (SSIM) and Feature Similarity Index Measure (FSIM) as a segmentation quality metrics to quantify the impact of segmentation with different algorithms.

PSNR is a metric used to measure the quality of reconstruction of a signal, especially in the context of image segmentation and compression. In this paper, it quantifies the amount of noise introduced during the segmentation process compared to the original images. The formula for calculating PSNR can be Mathematically defined as Eq (14):

$$\text{PSNR} = 20 \log_{10} \frac{2^n}{\text{MSE}} \tag{14}$$

where $2^n$ represents the maximum possible pixel value of image, for the 12-bit deep medical image here, the n is 12. The MSE indicates mean squared error between the original and segmented images. The larger the PSNR value, the less noise rest of the image, indicating good image segmentation quality.

SSIM is a method used to evaluate the perceived quality of reconstructed images. Unlike traditional metrics that rely solely on pixel-wise differences, SSIM takes into account the structural information and captures changes in structural similarities between the original and processed signals. The SSIM metric compares local patterns of pixel intensities rather than individual pixel values across multiple spatial scales. It is defined as the product of three key components of perception: luminance (l), contrast (c), and structure (s). The SSIM is expressed as Eq (15):

$$\text{SSIM}(x, y) = [l(x, y)]^{\alpha} \cdot [c(x, y)]^{\beta} \cdot [s(x, y)]^{\gamma} \tag{15}$$

in which the three modules l, c and s are defined respectively as follow:

$$l(x, y) = \frac{2\mu_x\mu_y + C_1}{\mu_x^2 + \mu_y^2 + C_1} \tag{16}$$

$$c(x, y) = \frac{2\delta_x\delta_y + C_2}{\delta_x^2 + \delta_y^2 + C_2} \tag{17}$$

$$s(x, y) = \frac{2\delta_{xy} + C_3}{\delta_x^2 + \delta_y^2 + C_3} \tag{18}$$

where $x$, $y$ represent the images before and after segmentation respectively. The parameters $\alpha$, $\beta$, $\gamma$ are constants that ensure the balance between luminance, contrast, and structure components, and in practical applications $\alpha = \beta = \gamma = 1$. $C_1$, $C_2$, $C_3$ is a constant and $C_3 = 0.5C_2$. $\mu_x$, $\mu_y$ are all pixels of the image, $\delta_x$, $\delta_y$ represents the standard deviation of the image pixel values, and $\delta_{xy}$ represents the covariance of the two images. The SSIM value ranges from 0 to 1, and a higher SSIM value closer to 1 indicates a better segmentation quality, suggesting that the processed image closely resembles the original in terms of structure and perception [47].

FSIM is a metric designed to evaluate the quality of images by assessing the similarity between the structural information of an original image and a segmentation image. During the calculation, FSIM considers both phase consistency feature (PC) and gradient feature (GM), in which PC can portray the local structure of an image and GM is used to extract the changes in an image [48].

The similarity of PC is calculated as Eq (19):

$$S_{PC}(x) = \frac{2PC_1(x) \cdot PC_2(x) + T_1}{PC_1^2(x) + PC_2^2(x) + T_1} \tag{20}$$

The similarity of GM is calculated as Eq (20):

$$S_{GM}(x) = \frac{2GM_1(x) \cdot GM_2(x) + T_2}{GM_1^2(x) + GM_2^2(x) + T_2} \tag{20}$$

The comprehensive similarity is calculated as Eq (21):

$$S_L(x) = S_{PC}(x) \cdot S_{GM}(x) \tag{21}$$

The formula for FSIM is as Eq (22):

$$\text{FSIM} = \frac{\sum_{x \in \Omega} S_L(x) \cdot PC_m(x)}{\sum_{x \in \Omega} PC_m(x)} \tag{22}$$

where $\alpha$, $\beta$ is generally taken as 1, $PC_m(x) = max(PC_1(x), PC_2(x))$.

The above 3 indicators are used to evaluate the quality of images segmented by different algorithms, and the numerical results of PSNR, SSIM and FSIM for image segmentation are shown in Tables 6–8, respectively.

Tables 6–8 show the qualify metrics of the tumor image segmented by different algorithms. Overall, the images segmented by the CIWP-PSO algorithm have obvious advantages in terms of PSNR, SSIM and FSIM. From Table 6 we can get that the CIWP-PSO obtains the highest FSNR values the most times (12), while other algorithms only obtained the optimal solution around 5 times.

Table 7 shows that for about half of the cases, CIWP-PSO achieved the best SSIM, which clearly far surpasses other algorithms. As for feature similarity, all algorithms achieved the best FSIM values between 3 to 5 times, except for PSO and CWIP-PSO, and it is easy to see that PSO and CWIP-PSO exerted the similar degree of influence on the image feature during the thresholding segmentation, but CIWP-PSO still has certain advantages.

In summary, it can be concluded that the image thresholding segmentation method based on CIWP-PSO provides competitive results compared to its famous and novel counterparts. The CIWP-PSO has more excellent multilevel thresholding segmentation ability for high-bit-depth images, which not only can obtain higher Kapur entropy in the image segmentation process, but also the images segmented by CIWP-PSO have higher quality in terms of FSNR, SSIM and FSIM.

**Table 6. Results of the PSNR measure for all algorithms.**

| Image id | nTh | SSA | PSO | GA | QPSO | PSOH | CIWP-PSO |
|---|---|---|---|---|---|---|---|
| Image a | 4 | 24.5956 | 24.4424 | 24.0684 | 24.8832 | 24.438 | **24.8856** |
| | 6 | 26.3348 | 28.049 | 27.8483 | 26.2756 | 26.2291 | **28.4368** |
| | 8 | 28.7647 | 29.3914 | 26.5176 | 29.9011 | **30.0931** | 29.3003 |
| Image b | 4 | 23.3883 | **25.2679** | 24.113 | 25.2565 | **25.2679** | **25.2679** |
| | 6 | 26.2105 | 27.063 | 25.2804 | 27.4936 | 26.7867 | **27.5584** |
| | 8 | 28.1319 | 30.3192 | 26.7587 | 29.8971 | 29.9732 | **30.5169** |
| Image c | 4 | 23.6106 | 24.2259 | 24.5974 | **24.6018** | 24.2281 | 24.2281 |
| | 6 | 26.6892 | 27.9431 | 24.437 | 28.0154 | **28.0165** | 27.886 |
| | 8 | 27.5465 | 28.8074 | 26.0483 | 28.8971 | 30.2043 | **30.254** |
| Image d | 4 | 21.5142 | 29.0857 | 29.4623 | **29.5618** | 29.0933 | 29.1042 |
| | 6 | 30.4084 | 32.0062 | 31.6368 | 32.0213 | 32.0611 | **32.0717** |
| | 8 | 28.9961 | 32.6642 | 30.9393 | 32.4456 | 32.517 | **34.3149** |
| Image e | 4 | **24.5849** | 24.4971 | 21.887 | 24.0198 | 24.5157 | 24.3406 |
| | 6 | 24.8053 | 25.6286 | 26.7176 | **27.8756** | 25.6796 | 27.7922 |
| | 8 | 23.7975 | **28.8294** | 26.3156 | 28.2873 | 26.2406 | 28.7917 |
| Image f | 4 | 20.7115 | 24.0274 | 21.9493 | 23.9028 | 18.0608 | **24.1172** |
| | 6 | **26.3088** | 26.2045 | 25.905 | 26.2368 | 26.3075 | 26.246 |
| | 8 | 23.2794 | 28.7626 | 26.5044 | 25.9082 | 28.1359 | **29.2011** |
| Image g | 4 | 20.9483 | 23.0312 | 21.1176 | **24.2066** | 22.9989 | 23.0009 |
| | 6 | 20.5896 | **26.9839** | 24.9636 | 26.7805 | 26.8121 | 26.9817 |
| | 8 | 24.7884 | 28.2808 | 23.8363 | 28.2901 | **28.9912** | 28.7103 |
| Image h | 4 | 22.4163 | 22.631 | 21.5582 | 20.9833 | 22.6158 | **22.7016** |
| | 6 | 22.7711 | **26.5491** | 24.9036 | 25.1744 | 24.4323 | 26.4835 |
| | 8 | 27.0744 | 26.4006 | 26.8735 | 27.3181 | **28.4173** | 27.4012 |
| Image i | 4 | 21.3263 | 21.6911 | **23.9697** | 21.0206 | 21.8994 | 21.4545 |
| | 6 | 22.326 | 27.2714 | 24.3483 | 27.4192 | **27.3705** | 27.3524 |
| | 8 | 21.2451 | 29.4433 | 28.4556 | 29.3211 | 29.6301 | **29.6972** |

## 5 Conclusions and future directions

In response to the insufficient performance of existing thresholding segmentation algorithms for high-bit-depth medical image, this paper proposes a multi-level threshold segmentation algorithm based on improved CIWP-PSO, which is a PSO variant improved by hybrid strategy. To adapt to the large search range brought by high-bit depth, some particles need to actively escape from the local optimum. Therefore, we constructed the particle swarm as a pyramid structure with three levels according to the fitness value, and the worst tier performs the adversarial learning strategy to explore as wide a search range as possible. In addition, to greatly improve convergence efficiency and accuracy, a pair of complementary inertia weight is introduced to guide individuals to search for optimal thresholds within a wider range. Then the Kapur entropy of images is used as the fitness function to search the optimal segmentation thresholds precisely. To validate the segmentation ability of high-bit depth images targeted by this proposed CIWP-PSO, the convergence performance is compared on 9 benchmark images and medical MRI image with 12-bit depth with its famous and excellent counterparts, such GWO, PSO, SSA and some PSO variants. The experimental results show that the multilevel thresholding segmentation algorithm based on CIWP-PSO outperforms the segmentation algorithms based on other SIOAs in the convergence accuracy. To evaluate the impact of

**Table 7. Results of the SSIM measure for all algorithms.**

| Image id | nTh | SSA | PSO | GA | QPSO | PSOH | CIWP-PSO |
|---|---|---|---|---|---|---|---|
| Image a | 4 | 0.694596 | 0.66725 | 0.691994 | 0.693592 | 0.667901 | **0.711307** |
| | 6 | 0.720311 | 0.751242 | 0.744642 | 0.735561 | 0.727089 | **0.75532** |
| | 8 | 0.761517 | 0.762948 | 0.763223 | 0.770233 | **0.775225** | 0.763491 |
| Image b | 4 | 0.622866 | 0.650561 | 0.639527 | 0.640713 | 0.650561 | **0.652481** |
| | 6 | 0.64724 | 0.663727 | 0.651395 | 0.679216 | **0.678423** | 0.664423 |
| | 8 | 0.677471 | 0.696654 | 0.646472 | 0.687862 | 0.693686 | **0.701726** |
| Image c | 4 | 0.638435 | 0.64439 | 0.628664 | 0.643562 | **0.646514** | 0.644272 |
| | 6 | 0.657739 | 0.683289 | 0.657381 | 0.685791 | **0.689769** | 0.686394 |
| | 8 | 0.692291 | 0.689734 | 0.646014 | 0.699232 | 0.69076 | **0.703986** |
| Image d | 4 | 0.539514 | 0.692611 | **0.693853** | 0.691846 | 0.688003 | 0.693162 |
| | 6 | 0.687932 | **0.716193** | 0.710888 | 0.715883 | 0.713196 | 0.715679 |
| | 8 | 0.689878 | 0.722963 | 0.70441 | 0.723154 | 0.706841 | **0.725679** |
| Image e | 4 | 0.70076 | 0.710189 | 0.664628 | 0.710931 | 0.710225 | **0.710762** |
| | 6 | 0.716324 | 0.72145 | 0.716107 | 0.737548 | 0.741241 | **0.743791** |
| | 8 | 0.70452 | 0.750253 | 0.748395 | 0.733484 | 0.74197 | **0.75241** |
| Image f | 4 | 0.489979 | 0.542575 | 0.457057 | 0.480349 | 0.537305 | **0.546936** |
| | 6 | 0.620898 | **0.644896** | 0.57903 | 0.643473 | 0.641281 | 0.64478 |
| | 8 | 0.590216 | 0.671775 | 0.609616 | 0.672342 | 0.528978 | **0.69312** |
| Image g | 4 | 0.601838 | 0.639844 | 0.613375 | 0.642374 | **0.657518** | 0.639888 |
| | 6 | 0.587923 | 0.694489 | 0.654115 | 0.692234 | 0.693441 | **0.694551** |
| | 8 | 0.650846 | 0.713883 | 0.669745 | **0.718324** | 0.716355 | 0.718261 |
| Image h | 4 | 0.564432 | 0.574259 | 0.576743 | 0.568923 | 0.369055 | **0.579174** |
| | 6 | 0.575382 | 0.674987 | 0.613475 | 0.647632 | 0.634147 | **0.677113** |
| | 8 | 0.677199 | 0.688304 | 0.658582 | **0.689234** | 0.682787 | 0.68193 |
| Image i | 4 | 0.539143 | 0.550148 | **0.636964** | 0.592342 | 0.33265 | 0.542314 |
| | 6 | 0.649444 | 0.714793 | 0.641287 | 0.713432 | **0.716685** | 0.715605 |
| | 8 | 0.621893 | 0.740898 | 0.70444 | 0.743243 | 0.742585 | **0.744294** |

segmentation on images, we calculated the FSNR, SSIM and FSIM of the two images before and after segmentation under different algorithms, and we conclude that CIWP-PSO not only has stronger image segmentation capabilities, but also has higher image segmentation quality in terms of PSNR and SSIM.

Segmentation of image is a foundational process with numerous medical applications, and it can be regarded as an upstream task of medical image processing, which prepares for downstream image analysis. Therefore, the quality of downstream tasks is another important criterion for evaluating performance of thresholding segmentation algorithms. In the future experiment, target extraction of brain tumor images will be used as the downstream task, and the target areas of the final images processed by the morphological operations will be compared to highlight the superiority of the CIWP-PSO.

As scope for further studies, the proposed CIWP-PSO can be applied to segment other kinds of images, such as underwater and remote sensing images, and we will make specific improvements to the segmentation algorithm according to the characteristics of different kinds of images. In addition, the other objective functions can be employed in the CIWP-PSO for searching the optimal thresholds.

**Table 8. Results of the FSIM measure for all algorithms.**

| Image id | nTh | SSA | PSO | GA | QPSO | PSOH | CIWP-PSO |
|---|---|---|---|---|---|---|---|
| Image a | 4 | 0.51118 | 0.498133 | 0.479006 | 0.502131 | 0.515625 | **0.519682** |
| | 6 | 0.550385 | 0.574165 | 0.569204 | 0.571233 | 0.559974 | **0.587774** |
| | 8 | 0.605582 | **0.624349** | 0.563663 | 0.615234 | 0.61459 | 0.604338 |
| Image b | 4 | 0.404999 | **0.445692** | 0.424475 | **0.445692** | 0.430914 | **0.445692** |
| | 6 | 0.496854 | 0.509182 | 0.468541 | 0.512343 | **0.526501** | 0.503088 |
| | 8 | 0.513179 | **0.585093** | 0.517067 | 0.573214 | 0.553769 | 0.583868 |
| Image c | 4 | 0.396279 | 0.398926 | **0.417281** | 0.409412 | 0.400722 | 0.407108 |
| | 6 | 0.488097 | 0.489188 | 0.425551 | 0.458765 | 0.489894 | **0.490643** |
| | 8 | 0.490016 | 0.528547 | 0.497866 | **0.534532** | 0.517416 | 0.543275 |
| Image d | 4 | 0.371156 | 0.414142 | 0.414967 | 0.434254 | **0.440542** | 0.413704 |
| | 6 | 0.464828 | 0.492112 | **0.50281** | 0.484352 | 0.491967 | 0.488047 |
| | 8 | 0.470352 | 0.521157 | 0.473794 | 0.534525 | 0.519268 | **0.548054** |
| Image e | 4 | 0.503415 | 0.473258 | 0.482224 | 0.523414 | **0.516818** | 0.505943 |
| | 6 | 0.524406 | 0.527902 | 0.552767 | 0.554235 | **0.582004** | 0.569329 |
| | 8 | 0.533025 | 0.582276 | 0.568359 | 0.582534 | 0.590937 | **0.597466** |
| Image f | 4 | 0.41903 | **0.497257** | 0.421261 | 0.434524 | 0.482861 | 0.492476 |
| | 6 | 0.539953 | 0.530858 | **0.558701** | 0.543523 | 0.548046 | 0.525067 |
| | 8 | 0.479105 | 0.613659 | 0.561443 | 0.592345 | 0.405194 | **0.619356** |
| Image g | 4 | 0.418193 | 0.461028 | 0.442636 | 0.464525 | **0.489799** | 0.448729 |
| | 6 | 0.457745 | 0.559622 | 0.514171 | 0.545635 | 0.546131 | **0.56006** |
| | 8 | 0.537376 | 0.587383 | 0.520971 | **0.608456** | 0.595611 | 0.604149 |
| Image h | 4 | 0.48775 | 0.490482 | 0.484315 | 0.484535 | 0.406806 | **0.496913** |
| | 6 | 0.516663 | **0.57744** | 0.551677 | 0.554366 | 0.556101 | 0.564578 |
| | 8 | 0.589432 | 0.573465 | 0.608449 | **0.645636** | 0.586851 | 0.608605 |
| Image i | 4 | 0.432857 | **0.448682** | 0.443268 | 0.447867 | 0.334914 | 0.43102 |
| | 6 | 0.452565 | **0.548361** | 0.509525 | 0.519435 | 0.544373 | 0.527984 |
| | 8 | 0.435531 | 0.614393 | 0.590084 | 0.619455 | 0.600577 | **0.625999** |

## Author Contributions

**Conceptualization:** Jianmin Hu.

**Data curation:** Jianmin Hu.

**Formal analysis:** Jianmin Hu.

**Investigation:** Jianmin Hu.

**Project administration:** Jiaqi Ma.

**Resources:** Jiaqi Ma.

**Software:** Jiaqi Ma.

**Supervision:** Jiaqi Ma.

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
