## [Decision Letter · Decision Letter 0]

23 Apr 2024

PONE-D-24-11887An improved particle swarm optimization for multilevel thresholding medical image segmentationPLOS ONE

Dear Dr. Ma,

Thank you for submitting your manuscript to PLOS ONE. After careful consideration, we feel that it has merit but does not fully meet PLOS ONE’s publication criteria as it currently stands. Therefore, we invite you to submit a revised version of the manuscript that addresses the points raised during the review process.

We look forward to receiving your revised manuscript.

Kind regards,

Sunil Pathak, Ph.D.

Academic Editor

PLOS ONE

Journal Requirements:

2. Please note that PLOS ONE has specific guidelines on code sharing for submissions in which author-generated code underpins the findings in the manuscript. In these cases, all author-generated code must be made available without restrictions upon publication of the work. 

Please review our guidelines at https://journals.plos.org/plosone/s/materials-and-software-sharing#loc-sharing-code and ensure that your code is shared in a way that follows best practice and facilitates reproducibility and reuse.

3. Thank you for stating the following in your Competing Interests section: "NO authors have competing interests"

4. We note that your Data Availability Statement is currently as follows: 

"All relevant data are within the manuscript and its Supporting Information files."

5. Please ensure that you refer to Figures 1 and 2 in your text as, if accepted, production will need this reference to link the reader to the figure.

6. We note that Figure 2 in your submission contain copyrighted images. All PLOS content is published under the Creative Commons Attribution License (CC BY 4.0), which means that the manuscript, images, and Supporting Information files will be freely available online, and any third party is permitted to access, download, copy, distribute, and use these materials in any way, even commercially, with proper attribution. For more information, see our copyright guidelines: http://journals.plos.org/plosone/s/licenses-and-copyright.

1) You may seek permission from the original copyright holder of Figure 2 to publish the content specifically under the CC BY 4.0 license. 

2) If you are unable to obtain permission from the original copyright holder to publish these figures under the CC BY 4.0 license or if the copyright holder’s requirements are incompatible with the CC BY 4.0 license, please either i) remove the figure or ii) supply a replacement figure that complies with the CC BY 4.0 license. Please check copyright information on all replacement figures and update the figure caption with source information. 

If applicable, please specify in the figure caption text when a figure is similar but not identical to the original image and is therefore for illustrative purposes only.

Reviewers' comments:

Reviewer's Responses to Questions

**Comments to the Author**

1. Is the manuscript technically sound, and do the data support the conclusions?

Reviewer #1: Yes

Reviewer #2: Partly

2. Has the statistical analysis been performed appropriately and rigorously? 

Reviewer #1: Yes

Reviewer #2: No

3. Have the authors made all data underlying the findings in their manuscript fully available?

Reviewer #1: Yes

Reviewer #2: No

4. Is the manuscript presented in an intelligible fashion and written in standard English?

Reviewer #1: Yes

Reviewer #2: No

5. Review Comments to the Author

Reviewer #1: 1)In the last line of the abstract be specific about the image quality (fitness value).

2)In literature review at least add 2/3 papers from 2024 also.

3)In reference use same format. In reference number 11, 19, 29, 32, 35, 39, 40, 46, I suggest you to use et. al format only.

4)Technical analysis is impressive but in your result section compare your results with latest papers from 2024.

5)I suggest you to use at least 15 benchmark images for comparison, you have used only 9 benchmark images.

6)Proposed method is well explained.

Reviewer #2: The ovearll quality of the paper needs improvement. Please answer the following comments:

1. How does the segmentation process described in Equation (3) adapt to different thresholds to divide an image into multiple regions?

2. Can you explain the significance of the objective function f(I,t1,t2,t3....tn) in thresholding image segmentation, and how does it relate to evaluating the quality of the segmented image?

3.Describe the Otsu method and the Kapur entropy method as mentioned in the text. How do they differ in their approach to image segmentation evaluation?

4.What is the rationale behind selecting the Kapur entropy method over the Otsu method for optimization in CIWP-PSO?

5.Explain the Random Opposition Learning (ROL) strategy and its role in the optimization process. How does it contribute to overcoming local optima?

6.Describe the hierarchical structure proposed in the improved CIWP-PSO. How does this structure facilitate the application of the ROL strategy?

7.What is the role of the complementary inertia weight in the PSO algorithm, and how does it address the limitations of traditional inertia weights?

8.Explain the rationale behind using a pair of complementary inertia weights based on fitness value. How does this approach adapt to the changes in fitness and iteration times?

9.How does the inverse Sigmoid function contribute to defining the behavior of inertia weight and reverse weight in the proposed algorithm?

10.Can you describe the overall impact of the proposed improvements on the CIWP-PSO algorithm's performance in optimizing thresholding image segmentation?

Kindly cite the below paper for better understanding the concept to Readers:

Wang, Baojun, et al. "Vibration compensation for railway track displacement monitoring system using biomedical image processing concept." The Journal of Engineering 2022.11 (2022): 1076-1085.

Srinivas, T.A.S., Yadav, R., Gowri, V., Chandraprabha, K., Ponnusamy, S. and Mavaluru, D., 2024. Development and implementation of unmanned vehicles through artificial intelligence involving communication system with sensors and control parameters. Measurement: Sensors, p.101136.

D. Singh, A. S. Keerthi Nayani, M. Sundar Rajan, R. Yadav, J. Alanya-Beltran and M. K. Chakravarthi, "Implementation of Virual Instrumentation for Signal Acquisition and Processing," 2022 International Conference on Innovative Computing, Intelligent Communication and Smart Electrical Systems (ICSES), Chennai, India, 2022, pp. 1-4, doi: 10.1109/ICSES55317.2022.9914143.

Yadav, R., & Tripathi, A. (2022). Machine learning theory and methods. In Intelligent system algorithms and applications in science and technology (pp. 101-116). Apple Academic Press.

6. PLOS authors have the option to publish the peer review history of their article (what does this mean?). If published, this will include your full peer review and any attached files.

Reviewer #1: **Yes: **Dr. Katyayani Kashyap

Reviewer #2: No

---

## [Author Response · Author response to Decision Letter 0]

6 Jun 2024

Dear editor and reviewers,

Thank you for offering us an opportunity to improve the quality of our submitted manuscript (An improved particle swarm optimization for multilevel thresholding medical image segmentation). We appreciated the editor and reviewers’ constructive and insightful comments very much. 

In this revision, we have addressed all these comments. We hope the revised manuscript has now met the publication standard of your journal. These changes are summarized below following a point-by-point response to editor and reviewers’ comments.

To Editor:

 Requirement 1: Please ensure that your manuscript meets PLOS ONE's style requirements, including those for file naming.

I have revised the formatting of the title, authors, affiliations, and body of this manuscript based on the formatting document you provided.

 Requirement 2: All author-generated code must be made available without restrictions upon publication of the work.

The relevant code of the paper has been submitted to the open-source code library. The code library address is as follows: https://github.com/JiaQiMark/CIWP-PSO.

 Requirement 3：Please complete your Competing Interests on the online submission form to state any Competing Interests.

The authors have declared that no competing interests exist.

 Requirement 4: Please confirm at this time whether your submission contains all raw data required to replicate the results of your study.

The manuscript already contains most of the data, and the remaining data is stored in the public repository: https://github.com/JiaQiMark/CIWP-PSO/tree/main/result.

 Requirement 5: Please ensure that you refer to Figures 1 and 2 in your text as, if accepted, production will need this reference to link the reader to the figure.

This article does refer Figure 1, and the Figure 1 is already presented in the manuscript. Figure 2 and its related parts have been removed.

 Requirement 6: Present written permission from the copyright holder to publish these figures specifically under the CC BY 4.0 license.

Since we were unable to obtain the written permission form the copyright holder, we collected and released our own dataset, replacing the original images with our own images. 

We have published our own dataset at the https://github.com/JiaQiMark/Benchmark-images-for-threshilding-segmentation, and we promise that the dataset is published under the Creative Commons Attribution License (CC BY 4.0).

 Requirement 7: Please review your reference list to ensure that it is complete and correct.

I have reviewed my reference list to ensure that it is complete and correct in terms of content and format. I did not cite papers that have been retracted.

To Reviewer #1:

 In the last line of the abstract be specific about the image quality (fitness value).

Thank you very much for your comments. We have revised and added specific instructions for image quality in the last line of the abstract. The added part is marked with red font.

 In literature review at least add 2/3 papers from 2024 also.

Thank you very much for your comments. We have discussed 3 relevant papers from 2024 in the literature review, and added part is marked with red font.

 In reference use same format. In reference number 11, 19, 29, 32, 35, 39, 40, 46, I suggest you use et. al format only.

Thank you very much for your comments. The formatting of references we used previously was really confusing, and we have now revised the formatting of all references as requested by you and the editor.

 Technical analysis is impressive but in your result section compare your results with latest papers from 2024.

Thank you very much for your comments. It is indeed more convincing to compare our method with the latest papers from 2024. Therefore, in the experimental part, the QPSO, published in the paper from 2024, is introduced to compare with our method, which uses an improved strategy to optimize PSO and has good performance.

 I suggest you use at least 15 benchmark images for comparison, you have used only 9 benchmark images.

Thank you very much for your comment. The 9 benchmark images are indeed too few to illustrate the superiority of the CIWP-PWO, so we finally used 16 benchmark images to investigate the performance of the CIWP-PSO in thresholding segmentation.

 Proposed method is well explained.

Thank you for your approval.

To Reviewer #2:

 How does the segmentation process described in Equation (3) adapt to different thresholds to divide an image into multiple regions?

Thank you very much for your detailed comments. I indeed only explained the case of dual region in the description of Equation (3). Now I have added to the explanation of Equation (3) how the equation can segment the image into multiple regions using different thresholds, and the added part is marked with blue font in subsection 3.2.

 Can you explain the significance of the objective function f(I, t1, t2, t3, ...., tn) in thresholding image segmentation, and how does it relate to evaluating the quality of the segmented image?

This article does lack the introduction and explanation of the objective function in Section 3.2, which makes the appearance of objective function seem abrupt and difficult to understand. We have added a description of the importance of the objective function and how it judges the quality of threshold segmentation in Section 3.2, and the added part is marked with blue font.

 Describe the Otsu method and the Kapur entropy method as mentioned in the text. How do they differ in their approach to image segmentation evaluation?

In subsection 3.2, we did not give full introduction to the Otsu method and the Kapur entropy method. Now we have added full introduction to the Otsu method and the Kapur entropy and explained how the two approaches are fundamentally different, the added part is marked with blue font.

 What is the rationale behind selecting the Kapur entropy method over the Otsu method for optimization in CIWP-PSO?

The reasons we gave for choosing the Kapur entropy were indeed insufficient and unclear. Now we have given the rationale behind selecting the Kapur entropy method, and the added part is marked with blue font. The rationale behind selecting the Kapur entropy method over the Otsu method for optimization in CIWP-PSO is as follows.

The Otsu method aims to the maximum inter-class variance of the grayscale values of different segmented regions, and the inter-class variance tends to be maximum when the areas of each region are similar. Therefore, the Otsu method is good at processing images with similar areas of different segmented regions, and it is easy to segment images with targets of different sizes into regions with similar sizes. However, Kapur entropy method aims to maximize the entropy, which is not sensitive to the size of the divided regions based on the gray-level intra-class probability and can better retain small targets in the image.

We have placed the explanation of this part in Section 3.2, which is marked in blue font.

 Explain the Random Opposition Learning (ROL) strategy and its role in the optimization process. How does it contribute to overcoming local optima?

Thank you very much for your valuable comments. Subsection 3.3.1 only gives a rough introduction of ROL, which may cause readers to be confused about ROL. We have now added a more detailed explanation of ROL in subsection 3.3.1, and explained how ROL helps to escape from local optima by introducing the core idea of ROL, and the added part is marked with blue font.

 Describe the hierarchical structure proposed in the improved CIWP-PSO. How does this structure facilitate the application of the ROL strategy?

In order to describe the hierarchical structure of CIWP-PSO, we attached a pyramid structure figure of population and explained the hierarchical structure in detail based on the figure. Following closely, we explained how this structure facilitate the application of the ROL strategy, and the added part is marked with blue font.

 What is the role of the complementary inertia weight in the PSO algorithm, and how does it address the limitations of traditional inertia weights?

The complementary inertia weight plays a role in adjusting movement inertia and controlling the movement distance during the evolution of individuals in a population, as described in Equation (6).

The complementary inertia weight addressed the limitations of traditional inertia weights in the following aspect. Traditional inertia weight often only acts on the step length and can only control the moving distance, as described in the following formula:

P_{t+1}=P_t+\\omega(f)\\bullet∆p

The pair of complementary inertia weights uses the inverse complementary relationship between step length and the inertia of the initial position vector, and they can conveniently control both the step length and inertia, as described in the following formula, which provides individuals with a direction of convergence towards excellent individuals that is clearer and more precise than that provided by traditional inertia weight.

P_{t+1}=\\omega(f)\\bullet P_t+\\omega_{reverse}(f)\\bullet∆p

We have reorganized the relevant parts of the pair of complementary inertia weights according to your comments and introduced the role of the complementary inertia weight in the PSO algorithm, how does it address the limitations of traditional inertia weights, and how does this approach adapt to the changes in fitness and iteration times. The deleted parts are crossed out, and the added part is marked with blue font.

 Explain the rationale behind using a pair of complementary inertia weights based on fitness value. How does this approach adapt to the changes in fitness and iteration times?

The rationale behind using the pair of complementary inertia weights is that the pair of weights utilize the opposite relationship between step length and the inertia of the initial position vector, and adjust both the step length and the inertia, which helps to improve convergence efficiency.

The pair of complementary inertia weights only adapt to the changes in the fitness value. The more excellent fitness value makes the particle has greater motivation to stay in its original position and the poor fitness value makes the particle has more motivation to leave current position. The larger fitness is, the more excellent fitness is. Therefore, the inertia weight \\omega(f) increases as the fitness increases, and reversed inertia weight \\omega_{reverse}(f) decreases as the fitness increases.

This issue has been modified along with Comment 7, which is marked in blue font.

 How does the inverse Sigmoid function contribute to defining the behavior of inertia weight and reverse weight in the proposed algorithm?

The inverse Sigmoid function satisfies the two requirements for the pair of inertia weights:

 The slope of the inverse Sigmoid function is monotonic throughout the variation range, such that the inertia weight \\omega(f) is positively related to fitness and reverse weight \\omega_{reverse}(f) is negatively related to fitness.

 As shown in the following figure, the inverse Sigmoid function has the larger slope at the beginning and end to ensure that more weights are distributed around the median, which can amplify the influence of inertia weight to accommodate search over a wide range.

The above description has been added to Section 3.3.2, which is marked in blue font.

 Can you describe the overall impact of the proposed improvements on the CIWP-PSO algorithm's performance in optimizing thresholding image segmentation?

The overall impact of improvement strategies proposed by this paper on the CIWP-PSO in optimizing thresholding image segmentation is mainly reflected in the two aspects:

 Random opposite learning strategy helps the CIWP-PSO give up exploring the worst areas and instead explore more potential thresholds, increase population diversity, and escape from the local optimal thresholds.

 The pair of complementary inertia weights, using the reverse changes of step size and inertia, provide CIWP-PSO with efficient guidance and feedback on its journey to explore the optimal thresholds, which greatly improves the efficiency of searching for the optimal thresholds in high bit-depth images.

We placed this overall impact at the end of Section 3, which is marked in blue font.

Kindly cite the below paper for better understanding the concept to Readers

Thank you for the papers you provided. These papers are highly relevant to this article, and they are helpful for readers to understand we article and related important concepts, so we have cited them as references [2] [17] [18] and [21].

---

## [Editor Report · Decision Letter 1]

15 Jun 2024

An improved particle swarm optimization for multilevel thresholding medical image segmentation

PONE-D-24-11887R1

Dear Dr. Ma,

We’re pleased to inform you that your manuscript has been judged scientifically suitable for publication and will be formally accepted for publication once it meets all outstanding technical requirements.

Kind regards,

Sunil Pathak, Ph.D.

Academic Editor

PLOS ONE

Additional Editor Comments (optional):

Based on the comments received from reviewers, the revised manuscript is in the condition to accepted for possible publication. 
---

## [Editor Report · Acceptance letter]

17 Jun 2024

PONE-D-24-11887R1 

PLOS ONE

Dear Dr. Ma, 

I'm pleased to inform you that your manuscript has been deemed suitable for publication in PLOS ONE. Congratulations! Your manuscript is now being handed over to our production team.

Kind regards, 

on behalf of

Prof.(Dr.) Sunil Pathak 

Academic Editor

PLOS ONE